# HetL, HetR and PatS form a reaction-diffusion system to control pattern formation in the cyanobacterium *nostoc* PCC 7120

Xiaomei Xu[1], Véronique Risoul[1], Deborah Byrne[2], Stéphanie Champ[1], Badreddine Douzi[3], Amel Latifi[1]*

[1]Aix Marseille Univ, CNRS, LCB, Laboratoire de Chimie Bactérienne, Marseille, France; [2]Aix Marseille Univ, CNRS, Protein Expression Facility, Institut de Microbiologie de la Méditerranée, Marseille, France; [3]Université de Lorraine, INRAE, DynAMic, Nancy, France

**Abstract** Local activation and long-range inhibition are mechanisms conserved in self-organizing systems leading to biological patterns. A number of them involve the production by the developing cell of an inhibitory morphogen, but how this cell becomes immune to self-inhibition is rather unknown. Under combined nitrogen starvation, the multicellular cyanobacterium *Nostoc* PCC 7120 develops nitrogen-fixing heterocysts with a pattern of one heterocyst every 10–12 vegetative cells. Cell differentiation is regulated by HetR which activates the synthesis of its own inhibitory morphogens, diffusion of which establishes the differentiation pattern. Here, we show that HetR interacts with HetL at the same interface as PatS, and that this interaction is necessary to suppress inhibition and to differentiate heterocysts. *hetL* expression is induced under nitrogen-starvation and is activated by HetR, suggesting that HetL provides immunity to the heterocyst. This protective mechanism might be conserved in other differentiating cyanobacteria as HetL homologues are spread across the phylum.

*For correspondence:
latifi@imm.cnrs.fr

**Competing interests:** The authors declare that no competing interests exist.

## Introduction

Periodic patterning of iterative forms is one of the most common features observed in developmental processes across the living kingdom. Unraveling the rules governing pattern formation helps to elucidate the genetic basis of cell differentiation. A key contribution in theoretical biology was made by Alan Turing in 1952 who demonstrated that the reaction of two molecules with different diffusion rates generates regular patterns (*Turing, 1952*). Later on, Turing equations were adapted in a new mathematical model that emphasized conspicuous features of biological development, in particular local activation linked to autocatalysis and long range inhibition (*Gierer and Meinhardt, 1972*; *Meinhardt and Gierer, 1974*). Valuable experimental approaches showed the recurrence of this mechanism in various developmental behaviors, such as: spacing of leaves, tissue regeneration in Hydra, epidermis of insects, pigmentation in Zebra fish, embryogenesis in *Drosophila*, and also differentiation in some prokaryotes (for a review see *Schweisguth and Corson, 2019*). In all these developmental situations several aspects of the molecular interactions involved to generate patterning remain to be elucidated.

Among prokaryotes, several members of the Cyanobacteria phylum are capable of cell differentiation. The molecular basis of differentiation has been well documented for the cyanobacterium *Anabaena/Nostoc* PCC 7120 (referred herein as *Nostoc*). *Nostoc* is a diazotrophic strain which can differentiate into specific type of cells responsible for fixing atmospheric nitrogen. When combined

**eLife digest** Cyanobacteria are the only bacteria on Earth able to draw their energy directly from the sun in the same way that plants do. In addition, some strains are able to 'fix' the nitrogen present in the atmosphere: they can extract this gas and transform it into nitrogen-based compounds necessary for life. However, both processes cannot happen in a given cell at the same time.

A strain of cyanobacteria called Nostoc PCC 7120 can organise itself into long filaments of interconnected cells. Under certain conditions, one in every ten cells stops drawing its energy from the sun, and starts fixing atmospheric nitrogen instead. Exactly how the bacteria are able to 'count to ten' and organize themselves in such a precise pattern is still unclear.

Cells can communicate and establish patterns by exchanging molecular signals that switch on and off certain cell programs. For instance, a protein called HetR turns on the genetic program that allows cyanobacteria to fix nitrogen; on the other hand, a signal known as PatS binds to HetR and turns it off. Cells starting to specialise in fixing nitrogen produce both HetR and PatS, with the latter diffusing in surrounding cells and preventing them from extracting nitrogen.

However, it remained unclear how the nitrogen-fixing cell could ignore its own PatS signal and keep its HetR signal active. HetL – another protein produced by the future nitrogen-fixing cell – could potentially play this role, but how it acts was unknown.

Here, Xu et al. show that HetL cannot diffuse from one cell to the other, and that it binds to HetR at the same place than PatS does. When both PatS and HetL are present, they compete to attach to HetR, which stops PatS from turning off HetR and deactivating the nitrogen-fixing program.

Understanding how cyanobacteria fix nitrogen could help to develop new types of natural fertiliser. More generally, dissecting how these simple organisms can create patterns could help to grasp how patterning emerges in more complex creatures.

nitrogen is abundant *Nostoc* forms long filaments consisting of a single cell type. When the filaments of *Nostoc* are deprived of combined nitrogen, around 10% of the vegetative cells differentiate into heterocysts. These micro-oxic cells, which provide a suitable environment for $N_2$-fixation, are non-dividing and are distributed semi-regularly along the filaments. *Nostoc* differentiation follows therefore a one-dimensional pattern of heterocysts separated by 10 vegetative cells. Heterocysts are unable to undergo cell division, but as vegetative cells continue dividing, the filaments grow and new heterocysts form in the middle of the intervals between pre-existing heterocysts. Consequently, the pattern is dynamic and persists throughout the growth (*Kumar et al., 2010*; *Flores and Herrero, 2010*). The molecular signal inducing heterocyst differentiation is 2-oxoglutarate (2-OG), which accumulates in response to combined nitrogen starvation (*Laurent et al., 2005*). Among the various genes involved in the regulation of heterocyst formation and patterning (*Herrero et al., 2016*), the global regulator NtcA and the specific master regulator HetR are key transcriptional factors in the cascade resulting in heterocyst development. Upon combined nitrogen starvation, NtcA interacts with 2-OG and induces heterocyst differentiation by controlling, directly or indirectly, the expression of several genes, including *hetR* (*Herrero et al., 2004*; *Valladares et al., 2008*). HetR is essential for cell differentiation: its deletion abolishes differentiation, and its overexpression induces differentiation of multiple contiguous heterocysts under combined nitrogen-starvation and allows differentiation even under non-permissive conditions (*Buikema and Haselkorn, 1991*). It exists in different oligomeric states among which dimer and tetramer have been proposed to interact with DNA (*Huang et al., 2004*; *Valladares et al., 2016*). The HetR regulon includes hundreds of genes, which are either activated in response to nitrogen starvation or repressed in nitrogen-replete conditions (*Flaherty et al., 2014*; *Mitschke et al., 2011*; *Videau et al., 2014*). The structure of HetR shows a unique fold comprising three domains: helix-turn-helix, flap and hood; with the latter encompassing the binding site of PatS (*Kim et al., 2011*; *Hu et al., 2015*). A key event in pattern formation in *Nostoc* is the positive autoregulation of *hetR* occurring specifically in the differentiating cell (*Black et al., 1993*; *Rajagopalan and Callahan, 2010*) and the inhibition of HetR in neighboring cells by the product of *patS* (*Golden and Yoon, 2003*). The deletion of the *patS* gene leads to the formation of multiple contiguous heterocysts, and its overexpression inhibits the differentiation process and hence

induces a lethal phenotype under combined nitrogen starvation (*Yoon and Golden, 1998*). *patS* encodes a 13–17 amino acid peptide containing at its carboxy-terminal end a **RGSGR** pentapeptide (PatS-5) that interacts with HetR, which inhibits its DNA-binding activity (*Huang et al., 2004*; *Feldmann et al., 2011*) and blocks cell differentiation when added to culture medium (*Yoon and Golden, 1998*). The hexapeptide **ERGSGR** (PatS-6) derived from PatS is also able to interact with HetR and to inhibit its activity (*Hu et al., 2015*). A mutant strain producing a variant of HetR, R223W, that is no longer able to interact with PatS, forms multiple contiguous heterocysts (*Khudyakov and Golden, 2004*; *Hu et al., 2015*). PatS-dependent signals are considered to diffuse along the filament; inhibiting HetR in the cells adjacent to the heterocyst, so preventing them from differentiating (*Risser and Callahan, 2009*; *Corrales-Guerrero et al., 2013*). The **RGSGR** pentapeptide is also present in HetN; a protein required for the maintenance of the pattern (*Black and Wolk, 1994*; *Higa et al., 2012*; *Corrales-Guerrero et al., 2014*), and recently a third protein (PatX) containing a **RGTGR** motif, has been reported to inhibit heterocyst differentiation when overproduced in *Nostoc* (*Elhai and Khudyakov, 2018*). The transcription of *patX* is induced in the heterocyst and is under the control of NtcA (*Elhai and Khudyakov, 2018*).

The HetR/PatS regulatory loop fits the local activation/long range inhibition scheme that has been adapted from the Turing model explaining pattern formation (*Brown and Rutenberg, 2014*; *Turing, 1952*; *Turing, 1990*; *Gierer and Meinhardt, 1972*; *Figure 1A*). Furthermore, the action of HetN and PatX as inhibitory factors, the stochastically noisy expression of regulatory proteins (HetR and NtcA) among other features specific to cell differentiation of *Nostoc*, allow the emergence of more elaborated mathematical models that outline the principles governing pattern formation in cyanobacteria (*Di Patti et al., 2018*; *Muñoz-García and Ares, 2016*).

Because *Nostoc* is the simplest model to address development in a one-dimensional self-organizing system, valuable genetics and biochemical studies have provided an accurate picture of heterocyst patterning. Nevertheless, several questions are still unanswered and deserve investigation. In particular, how the differentiating cell, where *patS* is expressed at its highest level, becomes immune to self-inhibition is not fully understood yet. A genetic analysis of *patS* suggested that the seven amino acids at the N-terminus of PatS protect the producing cell from inhibition likely concomitantly with the export of the active form of PatS (*Corrales-Guerrero et al., 2013*). Intriguingly, if *patS* or *patS-5* are expressed specifically in the vegetative cells, or when PatS-5 is added to the culture medium, cell differentiation is abolished. However when produced in cells that have already initiated development (proheterocysts), PatS-5 is not able to inhibit differentiation (*Yoon and Golden, 2001*; *Wu et al., 2004*). These observations suggest that proheterocysts must acquire additional protection post-PatS export/processing.

The *hetL* gene (all3740) was unearthed in a genetic screen aiming to identify factors involved in PatS signaling (*Liu and Golden, 2002*). HetL is a single domain protein composed of 40 pentapeptides (A(D/N)LXX), adopting a right-handed quadrilateral beta helix typical of an Rfr-fold common to all pentapeptide repeats containing proteins (PRPs) (*Ni et al., 2009*). The ectopic expression of *hetL* in a background of *patS* overexpression restores the ability of the strain to differentiate heterocysts (*Liu and Golden, 2002*). *hetL* overexpression stimulates differentiation also when PatS-5 is added to the culture medium (*Liu and Golden, 2002*). Henceforth, HetL interferes with PatS inhibition but the molecular mechanism involved is unknown.

This study aims to further explore the function of HetL in PatS signaling. We show that HetL interacts with HetR at the same interface as PatS. This interaction is needed for HetR to escape PatS inhibition, and does not inhibit its DNA-binding activity. Analyzing *hetL* transcription, we found that this gene is induced shortly after initiation of cell differentiation and that HetR is required for its expression. Finally, we show that the expression of *hetL* in heterocysts, but not in vegetative cells, is necessary to counteract PatS inhibitory effect. We conclude that, by interacting with HetR, HetL interferes with PatS fixation and therefore provides immunity to the developing cell.

## Results

### HetL interacts with HetR without inhibiting its DNA-binding activity

To get further insights into HetL function, we wondered whether its activity would be mediated by its direct interaction with HetR. To test this, we used the bacterial two hybrid assay (BACTH), which

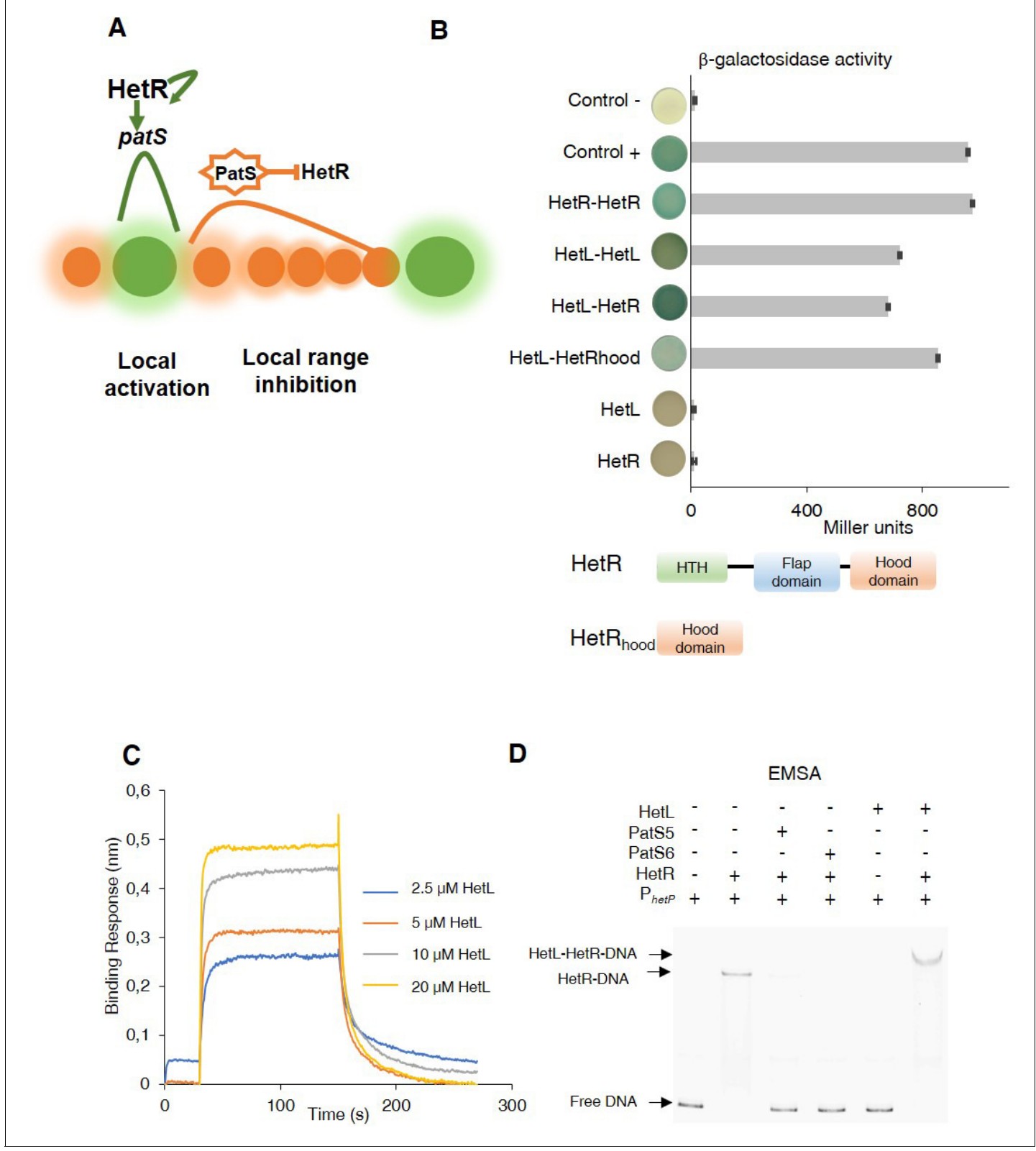

**Figure 1.** Patterning model and HetL/HetR interaction. (A) Self-organized patterning in *Nostoc* Heterocysts are presented in green. Vegetative cells in brown. The width of the shade around the cells represents the strength of the activation/inhibition. Processed PatS is framed by a star. (B) (*upper*) Bacterial two hybrid assay between HetL and HetR. BTH101 strain was transformed with pKT25-hetL and pUT18C-HetR, β-galactosidase activities were measured as described in section 'Materials and methods' and were expressed in Miller units. Strains producing the T18 with T25 (control -), T25-HetR

*Figure 1 continued on next page*

*Figure 1 continued*

with T18 (HetR) and T25 with T18-HetL (HetL) served as negative controls. Strains producing T18-Zip and T25-Zip served as positive control (control +). Error bars indicate standard deviation. The characteristics of the fusion proteins used in this assay are indicated below: HetR-HetR: T25-HetR/T18-HetR; HetL-HetL: T25-HetL/T18-HetL; HetL-HetR: T25-HetL/T18-HetR; HetL-HetRhood: T25-HetL/HetRhood-T18; HetL: T25-HetL/T18; HetR: T25/T18-HetR. (*lower*) Domains organization of HetR: Helix-turn-helix domain, flap domain and hood domain. (**C**) BLi assay between HetL and HetR. 2.2 μM of biotinylated HetR was loaded onto streptavidin biosensors. A 30 s baseline in PBS was performed before a 120 s association step with various concentrations of HetL at 2.5, 5, 10 and 20 μM followed by a 120 s dissociation step. Each curve represents the average of two experiments minus the control experiment. As a negative binding control, HetL 20 μM was added to the empty biosensor devoid of HetR. (**D**) EMSA assay of HetR (1 μM) with the *hetP* promoter (50 nM) in the presence of or PatS-5 or PatS-6 (1 μM) and HetL (4 μM). The *hetP* promoter incubated alone served as negative control (free DNA) and HetL plus DNA as a specific control for the binding activity of HetR.

The online version of this article includes the following source data and figure supplement(s) for figure 1:

**Source data 1.** Source data to *Figure 1B*.
**Source data 2.** Source data to *Figure 1C*.
**Figure supplement 1.** Purified HetL and HetR proteins.

is based on the reconstitution of adenylate cyclase (CyA) activity by two interacting proteins that bring the T18 and T25 domains of CyA into close proximity (*Karimova et al., 1998*). T18 and T25 domains were fused to HetR and HetL proteins at their N-terminal extremities, and the dimerization ability of HetR was used as an internal control for this assay. The data in *Figure 1B* show that HetL displays a strong interaction with HetR. Interestingly, it seems that the HetR-hood domain is sufficient to mediate the interaction of HetR to HetL. Furthermore, this experiment indicated that HetL is able to form dimers (or oligomers) (*Figure 1B*).

To confirm this interaction, we developed a BioLayer interferometry (BLi) assay. For this purpose, HetR and HetL proteins were produced and purified using affinity chromatography (*Figure 1—figure supplement 1*). HetR was biotinylated and immobilized on streptavidin biosensors as the ligand, while HetL was used as the analyte. Upon addition of HetL, a concentration-dependent association was recorded and decreased during the dissociation step corresponding to the washing of the sensor, indicating a direct interaction between HetL and HetR (*Figure 1C*). The estimated dissociation constant ($K_D$) of the HetR-HetL interaction was 6 μM. The interaction between HetR and HetL observed in the BACTH assay was thus confirmed by BLi.

As HetR acts by directly binding to promoters of a subset of its target genes, we tested if the interaction with HetL would impact its DNA binding activity. To this end, we conducted an electrophoretic mobility shift assay (EMSA) using the *hetP* promoter as a target (*Huang et al., 2004*; *Hu et al., 2015*). The previously reported ability of PatS-5 to inhibit HetR DNA-binding activity was used as a control (*Huang et al., 2004*; *Hu et al., 2015*). In the presence of HetL, HetR was still able to interact with the *hetP* promoter and the complex formed was higher than the one formed by HetR alone (*Figure 1D*). This result indicates that, contrary to PatS-5 binding, the interaction between the two proteins does not inhibit the DNA-binding activity of HetR.

## HetL and PatS interact with HetR at the same interface

Since HetL has been identified on the basis of counteracting PatS inhibition and as both PatS and HetL interact with HetR at its hood domain, we hypothesized that HetL could interact with HetR at the same interface as PatS, which would therefore interfere with its inhibiting action. To test this hypothesis, we took advantage of the fact that HetR-PatS interaction involves the Hood domain of HetR and that the residue R223 of HetR is required for this interaction. Interestingly, a variant of HetR bearing a R223W substitution lost the ability to interact with HetL (*Figure 2A*). HetR (R223W) was still able to form dimers (*Figure 2A*), which indicates that this variant is correctly folded. We conclude that the absence of interaction between HetR (R223W) and HetL indicates that this residue, in addition to being involved in the interaction with PatS-5/PatS-6, is also required for the interaction with HetL. For a deeper investigation of HetR-HetL interaction, we exploited the available structures of HetL and of the Hood domain of HetR to build interaction models (*Ni et al., 2009*; *Hu et al., 2015*). Interestingly, a group of four models among the top 10 clusters share similar orientations of HetR on HetL (*Figure 2—figure supplement 1*). Importantly, these models present interesting properties regarding the binding interface of HetR-Hood to HetL: (i) the residue R223 of HetR was involved in the binding interface with HetL (*Figure 2B*, *Figure 2—figure supplement 1*), (ii) the

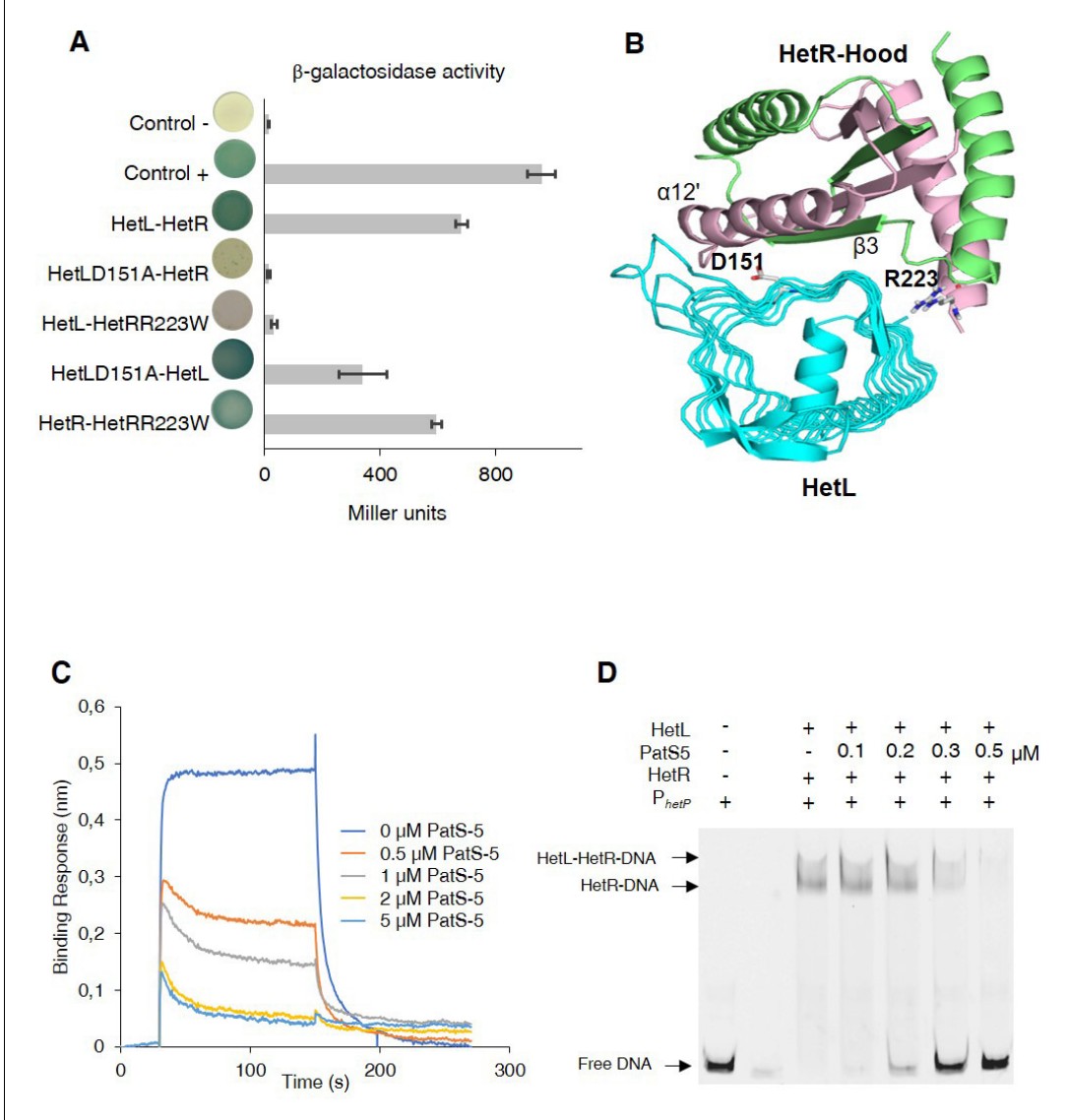

**Figure 2.** HetL-HetR interaction interface. (**A**) Bacterial two hybrid assay between HetL and HetR. β-galactosidase activities were measured as described in section 'Materials and methods' and were expressed in Miller units. Strains producing the T18 and T25 served as negative control (control -). Strains producing T18-Zip and T25-Zip served as positive control (control +). Error bars indicate standard deviation. The characteristics of the fusion proteins used in this assay are indicated below: HetL-HetR: T25-HetL/T18-HetR; HetLD151A-HetR: T25-HetLD151A/T18-HetR; HetL-HetRR223W: T25-HetL/T18-HetRR223W; HetLD151A-HetL: T25-HetLD151A/T18-HetL; HetR-HetRR223W: T25-HetR/T18-HetRR223W. (**B**) Model of HetR-Hood:HetL complex. Monomers of HetR-Hood dimer structure are presented in green and light pink. HetL is presented in cyan. D151 from HetL and R223 from HetR are indicated. (**C**) BLi assay between HetL and HetR in the presence of PatS-5. 10 µM of HetL was incubated 5 min with different concentrations of PatS at 0.5, 1, 2.5 and 5 µM before bringing HetL in contact with the bound HetR. Each curve represents the average of two experiments minus the control experiment. Control experiment HetL plus different concentrations of PatS loaded onto a biosensor devoid of HetR. (**D**) EMSA assay of HetL (2 µM) and HetR (1 µM) with the *hetP* promoter in the presence of PatS-5 at different concentrations. Note that, to distinguish the HetL/HetR-DNA and HetR-DNA complexes the concentration of HetL used in this experiment was lower than that shown in *Figure 1*.

The online version of this article includes the following source data and figure supplement(s) for figure 2:

**Source data 1.** Source data to *Figure 2A*.
**Source data 2.** Source data to *Figure 2C*.
**Figure supplement 1.** Best models obtained from docking simulations.
**Figure supplement 2.** BLi assay between HetL and HetR in the presence of PatS-6.
**Figure supplement 2—source data 1.** BLi assay between HetL and HetR in the presence of PatS-6.
**Figure supplement 3.** PatS-6 interferes with HetL-HetR interaction.

HetR interaction interface with HetL matches the one involved in PatS-5 interaction (*Figure 2B*). In the retained models, the HetR-Hood:HetL interaction is maintained by a large network of electrostatic interactions and hydrogen bonds. For HetR, the binding interface is composed of the surface exposed residues from strand β3 and α-helix α12′ (*Figure 2B*). In HetL, the large proportion of the binding interface includes the pentapeptides repeat localized in face three encompassed E[84]ADLT, K[104]ASLC, [124]QADLR, [149]YADLR, [169]RANFG and [197]YANLE. A small second binding interface is localized within face 2 and includes [40]ADLRQ and [145]ADLSY pentapeptide repeats. To gain further insight about the HetR:HetL interaction, we performed a site-directed mutagenesis to substitute the residue D151 from HetL to Alanine. Interestingly, this substitution impaired the binding of HetL to HetR as revealed by BACTH assay (*Figure 2A*). Furthermore, HetL D151A variant was still able to form dimers which is an important indication to rule out the possibility that the mutation impacted the fold of this variant (*Figure 2A*). Taken together, these results indicate that HetL and PatS-5 interact with HetR at the same interface, which implies that HetL and PatS must compete for the interaction with HetR. To check this assumption, the interaction between HetR and HetL was analyzed by BLi in the presence of increasing amounts of PatS-5. The data obtained clearly indicated that the association between HetR and HetL is impaired with the addition of PatS-5 in a dose-response manner, with a total inhibition effect obtained at a concentration of 5 µM (*Figure 2C*). Similar results were obtained with PatS-6 (*Figure 2—figure supplement 2*).

The effect of PatS on HetR-HetL interaction was further analyzed by two hybrid assays. For this, a synthetic operon constituted of *T18-hetL* and *patS-6* was constructed and used to question the interaction with HetR. While adenylate cyclase activity was restored in bacteria producing T25-HetR and T18-HetL fusions, the interaction between HetR and HetL was abolished when PatS-6 was produced along with HetL from the synthetic operon (*Figure 2—figure supplement 3*), which is a further demonstration that PatS and HetL share the same interaction site within HetR. The interference of PatS with the HetR-HetL complex was also observed in EMSA experiments where the addition of increasing amounts of PatS-5 abolished the binding of HetR-HetL to the *hetP* promoter (*Figure 2D*). Altogether, these data support the hypothesis of a competition between PatS and HetL for the interaction with HetR.

## HetL does not interact with PatS

Based on the data presented above, HetR acquired immunity against PatS inhibition can be explained by an exclusion of the inhibitor as a consequence of HetR-HetL interaction. However, an additional titration-based mechanism through a direct interaction between HetL and PatS cannot be ruled out. To test this possibility, Isothermal titration calorimetry (ITC) was used to analyze the possible interaction between HetL and PatS-5. This technique has been used previously to uncover the binding of PatS-5 to HetR (*Feldmann et al., 2011*). We have confirmed that HetR does indeed interact with PatS-5 with a dissociation constant ($K_D$) of 600 nM similar to that of 227 nM described in the literature with a stoichiomerty of 1:1. The data in *Figure 3* show that HetR (left panel) displays a reducing heat exchange when titrated with increasing amounts of Pats-5, indicating a saturation of the HetR sites with PatS-5. This reaction is an exothermic favorable reaction with an enthalpy of $-7.43 \pm 0.67$ ΔH kcal/mol. On the contrary, HetL (right panel) does not show a relative heat exchange upon binding to PatS-5. The interaction between HetR and PatS-5 observed in *Figure 3*, validates our technical experiments and revokes the possibility that HetL could titrate PatS-5 by a direct interaction.

## The interaction between HetL and HetR is required for HetR function in vivo

As *hetL* was discovered on the basis of its capacity, when overexpressed, to suppress the inhibitory effect of PatS, we used this approach to evaluate the impact of the interaction between HetR and HetL on the differentiation process. The wild-type version of *hetL* or the mutated gene encoding HetL (D151A) were expressed in *Nostoc* cells under the control of the *petE* promoter, and quantitative RT-PCR analyses were carried out to check the accurate overexpression of these genes upon induction. Results revealed a 30-fold higher expression of the *hetL* and the *hetL (D151A)* genes in the recombinant strains compared to the wild type (*Figure 4A*), indicating that the two versions of *hetL* are actually overexpressed. PatS effect was analyzed either by the overexpression of *patS* gene

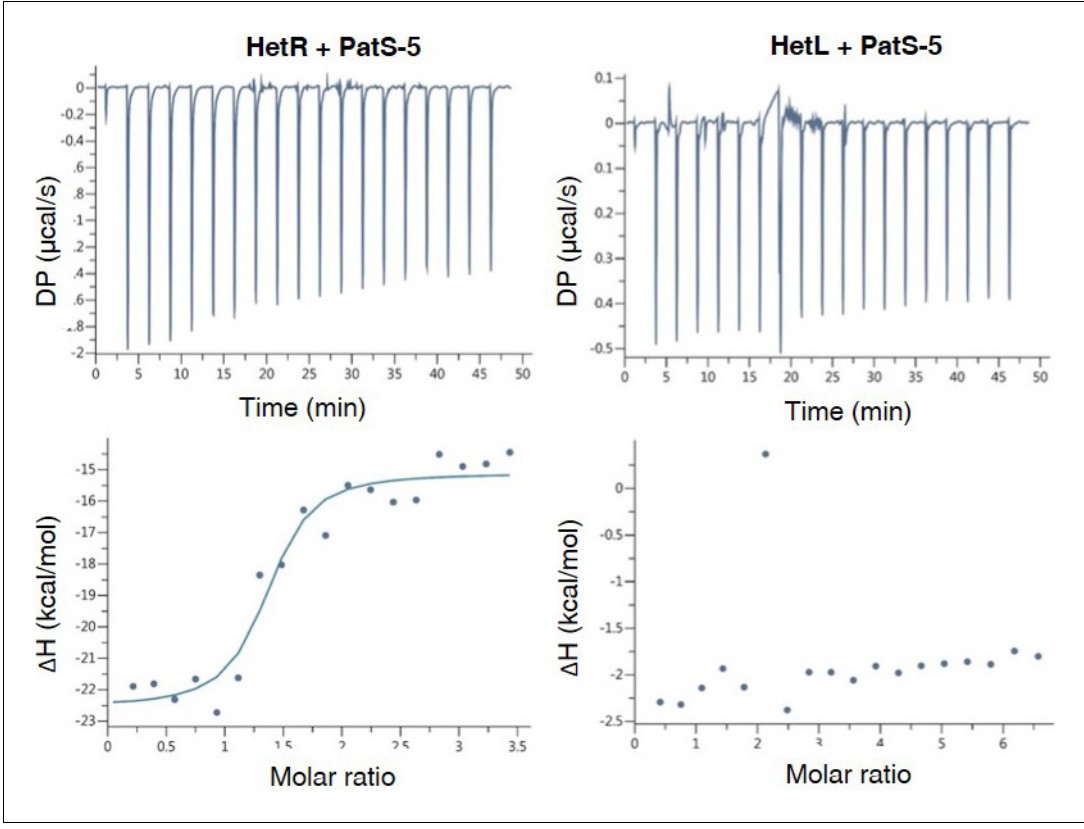

**Figure 3.** HetL does not interact with PatS-5. ITC experiment to detect HetL interaction with PatS-5. The right panel shows HetR and PatS-5 interaction. The left panel shows the non-interaction of HetL and PatS-5. The respective upper panels show heat exchange upon ligand titration and bottom panels show integrated data with binding isotherms (solid line) fitted to a single–site binding model. The constant heat dilution was removed before the integrated binding isotherms. The titrant PatS-5 (800 µM) was titrated into a cell containing 23 µM HetR at 25° C.

The online version of this article includes the following source data for figure 3:

**Source data 1.** Raw data for ITC experiments run with HetL (page 1) or HetR (page 2).

from the *petE* promoter or by the addition in the medium of PatS-5 (or PatS-6). The cultures were transferred into a nitrate-depleted medium during 48 hr to assess heterocyst development. In agreement with published data, the overproduction of PatS or the addition of PatS-5 inhibited the differentiation process, while the overexpression of *hetL* restored the ability of the PatS overproducing strain to form heterocysts (*Figure 4B*). HetL overproduction also allowed the ability of the strain to form heterocysts when PatS-5 was added to the culture (*Figure 4—figure supplement 1*). The percentage of the heterocysts formed in the *hetL* overexpressing strain was equal to that of the wild-type strain (*Table 1*). Interestingly, the recombinant strain overproducing HetL (D151A) was not able to form heterocysts when PatS was overproduced or upon addition of PatS-5 in the growth medium (*Figure 4B*, *Figure 4—figure supplement 1*). The interaction of HetL with HetR is therefore needed to counteract PatS inhibition. *hetL* expression in the heterocyst is required for providing HetR immunity against PatS inhibition *hetL* expression level was too low to be detected by Northern blot or fluorescent gene fusions (*Liu and Golden, 2002*). We therefore chose quantitative RT-PCR approach to analyze *hetL* transcription during the differentiation process. The *hetP* gene whose transcription is activated by HetR 8 hr after nitrogen step-down was used as an internal control for this experiment (*Mitschke et al., 2011*). RNAs were collected from the wild-type strain and the ΔhetR mutant at different times after nitrogen starvation and the transcript levels of the two genes were expressed relative to their amount at time zero. Results reveal that *hetP* expression was, as expected, strongly induced from 8 hr after nitrate depletion. Contrary to the 10-fold induction in the wild-type strain, the expression of *hetP* did not significantly increase in the ΔhetR strain which is consistent with the

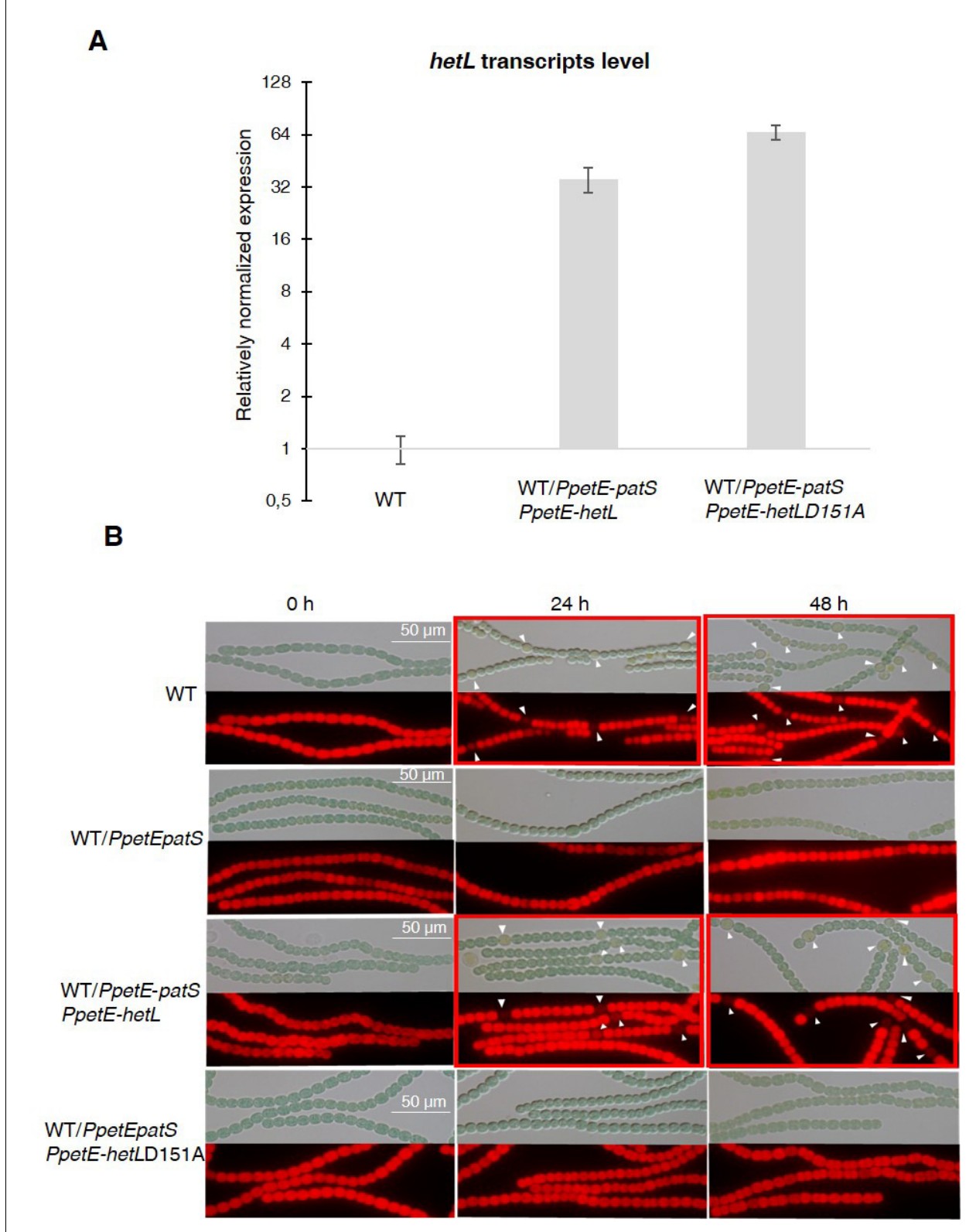

**Figure 4.** HetL-HetR interaction is required for heterocyst differentiation. (**A**) *hetL or hetL*D151A in the indicated strains was overexpressed from the *petE* promoter. Gene expression was induced with 3 µM copper. qRT-PCR experiments were undergone to determine *hetL or hetL*D151A expression level. Each sample was measured in triplicate and the standard deviation is indicated by error bars. (**B**) Microscopic bright field images (upper) and

*Figure 4 continued on next page*

*Figure 4 continued*

auto-fluorescence images (lower) of indicated *Nostoc* strains after nitrogen stepdown in addition of 3 µM copper. White arrows point to heterocysts. Images with heterocysts are framed in red.

The online version of this article includes the following source data and figure supplement(s) for figure 4:

**Source data 1.** Source data to *Figure 4A*.

**Figure supplement 1.** WT/P*petE-hetL* strain was able to form heterocysts in the presence of the PatS-5 peptide.

activation of this gene by HetR (*Figure 5A*). The *hetL* gene showed a similar transcription profile to that of *hetP*, yet its expression level was much lower. In the wild-type strain, a 3.5-fold increase of *hetL* transcripts was observed 8 hr after nitrogen step-down and was maintained up to 24 hr. In the Δ*hetR* mutant, no induction of *hetL* transcription was observed. The region including 500 nucleotides upstream of the start codon of *hetL*, which must include the promoter of this gene, was analyzed to probe for the two binding site consensus reported for HetR: the high-affinity consensus (GTAGGC-GA**GGG**GTCTAAC**CCC**TCATTACC), and the low-affinity one (GCTTAT**GGTGGGCAATGCCCACC**ATAATA) (*Videau et al., 2014*). None of these sequences are present in the *hetL* promoter. We concluded that, even if low, the transcription of *hetL* gene is induced early during the differentiation program and that HetR is required for *hetL* activation. The action of HetR on the promoter of *hetL* may be either indirect and mediated by another factor, or direct through a degenerated consensus which might explain the low transcription level of this gene.

From the results presented above, it can be deduced that HetL acts in the heterocyst. To further confirm this assumption, we expressed *hetL* either from the *rbcL* promoter, which is specific to vegetative cells, or from the *patS* promoter which is expressed in the heterocysts early after nitrogen stepdown. The ability of HetL to suppress heterocyst inhibition triggered by the addition of PatS-5 was analyzed. Results show that the strain expressing the P*patS-hetL* gene was able to form heterocysts even in the presence of PatS-5, but when expressed from the *rbcL* promoter, *hetL* was unable to prevent the inhibitory effect of PatS-6 (*Figure 5B*). This result is in favor of HetL acting specifically in the heterocyst.

## HetL provides immunity against PatX-derived pentapeptide

In addition to the RGSGR pentapeptide, heterocyst pattern formation has been recently shown to involve the H**RGTGR** peptide derived from the PatX protein (*Elhai and Khudyakov, 2018*). To further characterize HetL function, we wondered whether it would be involved in PatX signaling as well. Bli experiments showed that the HRGTGR peptide, like PatS-5, inhibited the interaction between HetR and HetL (*Figure 6A*). In addition, the strain expressing the P*petE-hetL* gene was able to form heterocysts even in the presence of the H**RGTGR** peptide (*Figure 6B*), while the overproduction of HetL (D151A) did not bypass PatX-6 inhibition since heterocysts were not observed (*Figure 6B*).

**Table 1.** Percentage of heterocysts formed by different strains used in this study after combined nitrogen starvation.

The number of the filaments analyzed was 60–100 in average.

| Strain and condition | % of heterocysts, 24H after nitrogen starvation | Mean interval between two heterocysts |
|---|---|---|
| Wild type | 9.1 ± 1.6 | 10.7 ± 1.3 |
| WT/P*petE-patS* | 0 | ND |
| WT and PatS-5 addition | 0 | ND |
| WT/P*petE-hetL*[D151A] | 10.5 ± 0.6 | 9.1 ± 1.6 |
| WT/P*petE-patS* P*petE-hetL* | 11.1 ± 2.4 | 6.1 ± 1.8 |
| WT/P*petE-patS* P*petE-hetL*[D151A] | 0 | ND |

The online version of this article includes the following source data for Table 1:

**Source data 1.** Raw data for heterocyst intervals and percentages.

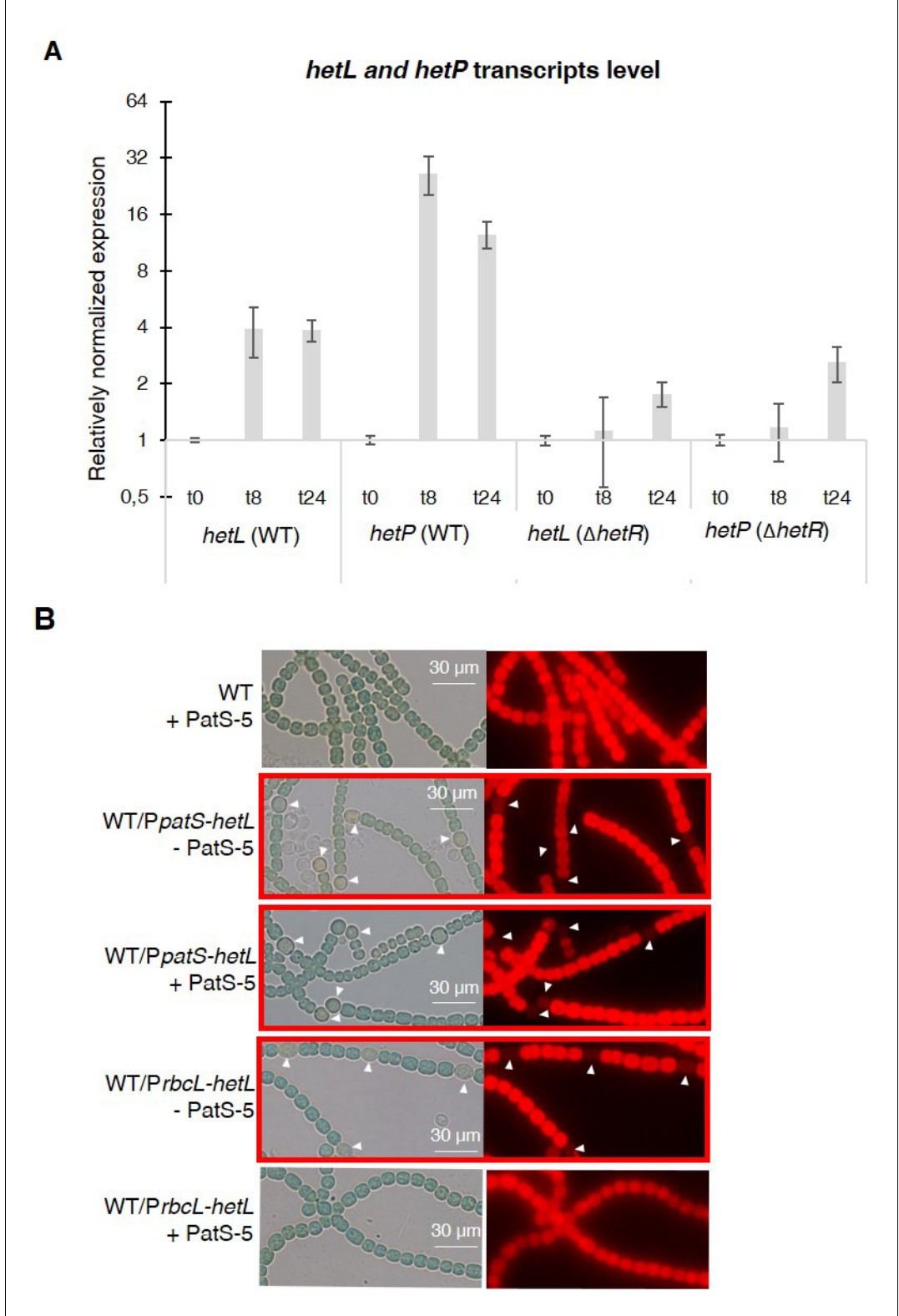

**Figure 5.** *hetL* expression is under the control of HetR. (**A**) *hetL* expression is induced in response to nitrogen starvation and is dependent on HetR. qRT-PCR was made to determine the *hetL* expression in WT and Δ*hetR* strains. *hetP* expression was determined as inner positive control. Each sample was measured in triplicate and the standard deviation is indicated by error bars (**B**) Effect of *hetL* expression from *patS* promoter or *rbcL* promoter. *Figure 5 continued on next page*

*Figure 5 continued*

Microscopic bright field images (left) and auto-fluorescence images (right) of indicated *Nostoc* strains after 24 hr of nitrogen stepdown are shown. White arrows point to heterocysts. Images with heterocysts are with red frames. The online version of this article includes the following source data for figure 5:

**Source data 1.** Source data to *Figure 4A*.

From these results, we conclude that HetL provides immunity for the developing cells against the two inhibitory peptides involved in pattern formation.

## HetR interacts with a pentapeptide repeat protein homolog to HetL

In addition to HetL, the genome of *Nostoc* contains 31 genes potentially coding for PRPs of various sizes. This large family includes proteins predicted to be located either in the cytoplasm, in the membrane or in the periplasm (*Ni et al., 2009*). Five of them display PRs among other domains, while the others, such HetL, are integrally formed by PR domains (*Supplementary file 1*). Given the considerable sequence identity shared by these PRPs (32% in average), predicting functional specificity based on sequence similarity is not possible. Because *hetL* mutant does not show any specific differentiation phenotype (*Liu and Golden, 2002*), we wondered if this could be due to a cross-complementation with another PRP. In this regard, we analyzed the capacity of HetR to interact with some HetL homologs. For this, we chose All3256 and All4303 because they share the closest features with HetL. They have a similar size (237, 268, 213 amino acids, respectively), a similar organization of the PRs domains, and the three are predicted to be cytosolic (*Figure 7A*). *Figure 7B* shows the results of a BACTH assay questioning the putative interaction of these proteins with HetR. Only All4303 displayed interaction with HetR, and even if this interaction is two-fold weaker than that of HetR-HetL it is significant compared to the negative control. This experiment indicates that at least one among the 31 PRPs is able to interact with HetR, which makes a cross-complementation of the *hetL* mutation with all4303, or another PRP coding gene a possible scenario. An important perspective of this work is to study the impact of the *hetL*-all4303 double mutation on the differentiation process.

## Discussion

Eighteen years ago, the *hetL* gene was discovered on the basis of its ability, when overexpressed, to bypass the inhibitory effect of *patS* overexpression on cell differentiation (*Liu and Golden, 2002*). In their conclusion, the authors of this study speculated that PatS and HetL might act by modulating HetR activity. This speculation reveals to be an accurate prediction as demonstrated by the results of the experiments presented in this manuscript, added to all the knowledge accumulated on HetR and PatS functions from previous valuable studies that are fundamental to our investigation. In particular, as the unique structural fold of HetR with its two exposed domains (flap and hood) was proposed to favor protein-protein interactions, we questioned the possible interaction of HetR with proteins involved in patterning. In this context, HetL was found to interact strongly with HetR without abolishing its DNA-binding activity, which suggests that HetR-HetL complex may be active regarding gene regulation in vivo (*Figure 1*). A possible mechanism to explain the role of HetL in PatS signaling is the titration by HetL of the inhibiting peptide. This hypothesis was ruled out since ITC assay did not show any interaction between HetL and PatS (*Figure 3*). Alternatively, a site-exclusion mechanism can be proposed for HetL. The observation that HetL and PatS-5 (or PatS-6) interact with HetR at the same interface is in agreement with this suggestion (*Figure 2B*). In Bli assays, increasing concentrations of PatS-5 interfered with HetR-HetL interaction, and in EMSA experiments addition of PatS-5 abolished the formation of HetR-HetL complex in a concentration-dependent manner (*Figure 2C–D*). In addition to confirming the in silico docking model predicting a same HetR-interaction interface for PatS and HetL, these data imply that the concentration of HetL in the (pro)heterocyst must be higher than the PatS peptide, which is plausible because (i) PatS peptide is diffusible (*Wu et al., 2004*; *Yoon and Golden, 2001*), (ii) HetL features do not include any motif or domain that predicts a putative translocation/secretion or association with membranes that could decrease its amount in the producing cell. Moreover, as *hetL* transcription is activated by HetR (*Figure 4*), HetL protein is likely to accumulate in the cytoplasm of differentiating cells. The diffusion of the inhibiting peptides (derived from PatS and HetN) along the filament has been shown to occur through septal junctions

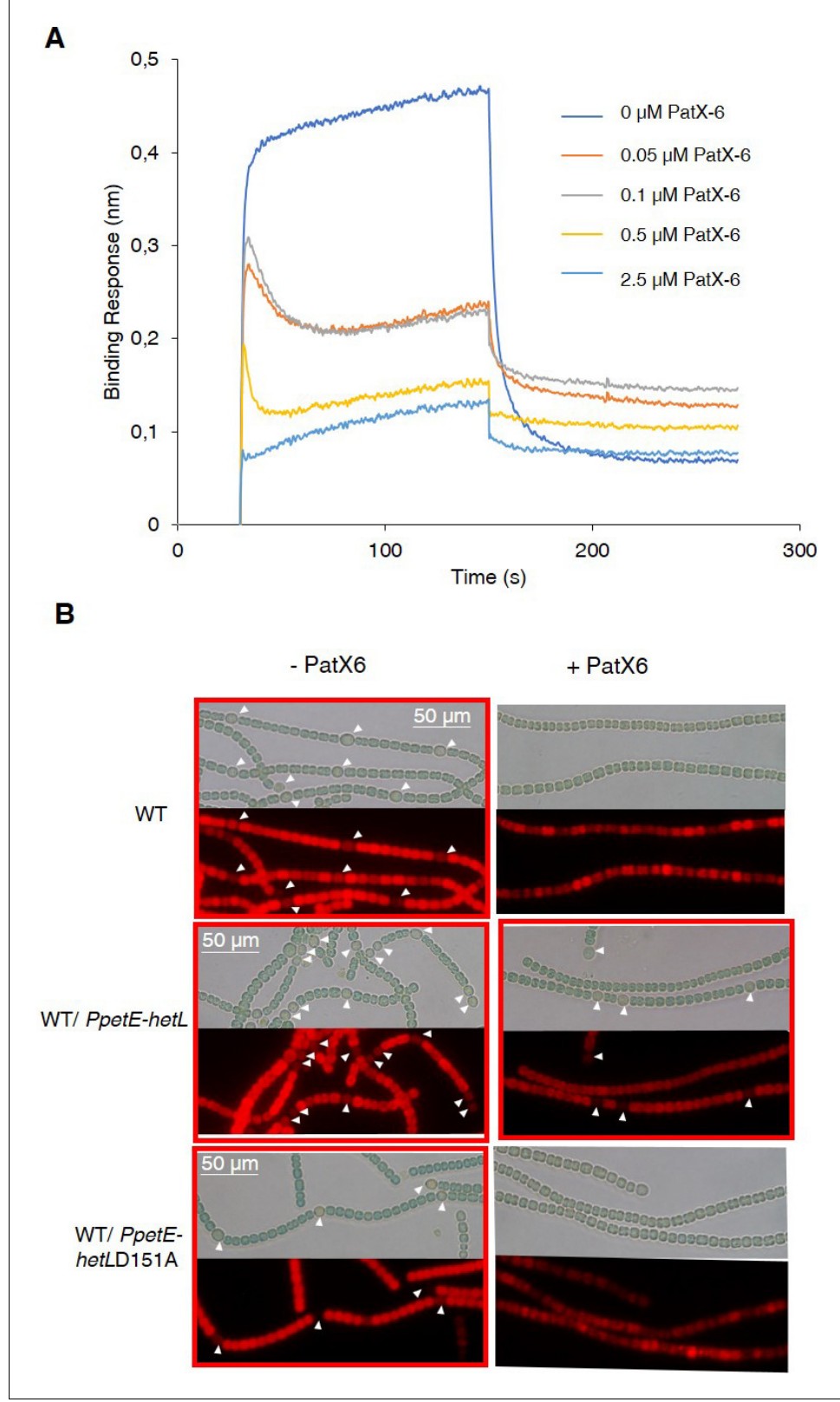

**Figure 6.** HetL provides protection against PatX-derived pentapeptide. (**A**) BLi assay between HetL and HetR in the presence of PatX-6 (HRGTGR). 10 µM of HetL was incubated 5 min with different concentrations of PatX6 at 0.05, 0.1, 0.5 and 2.5 µM before bringing HetL in contact with the bound HetR. Each curve represents the average of two experiments minus the control experiment. Control experiment HetL 10 µM loaded onto a biosensor

*Figure 6 continued on next page*

*Figure 6 continued*

devoid of HetR. (B) WT/P*petE-hetL* strain was able to form heterocysts even in the presence of the PatX-6 peptide. Microscopic bright field images (upper) and auto-fluorescence images (lower) of indicated *Nostoc* strains after 24 hr of nitrogen stepdown in addition of 3 µM copper are shown. White arrows point to heterocysts. Images with heterocysts are with red frames.

The online version of this article includes the following source data for figure 6:

**Source data 1.** Source data to *Figure 6A*.

(SJ) connecting adjacent cells (*Mariscal et al., 2016*). Electron cryotomography analysis showed that these SJ consist of tubes traversing the septal peptidoglycan (*Weiss et al., 2019*). Interestingly, the lumen of the tubes measures around 7 nm which is compatible with the diffusion of the inhibiting peptides but probably not with the transfer of HetL between cells. HetL protein is 3.5 nm wide and 6.5 nm length (*Ni et al., 2009*), which might suggest that a monomer could be transferred via the SJ. However, it is important to note that that HetL is capable of forming dimers (or multimers) (*Figure 1B*), the size of which is not consistent with transfer via the SJ. In addition, it is also possible that HetL surface charge might not be compatible with its transfer across the SJ tubes.

Alternatively, it can be assumed that in the (pro)heterocyst the affinity of HetR for HetL must be higher than that for PatS. Bli and ITC experiments showed that in vitro, the affinity of HetR for PatS is 10-fold higher than for HetL (compare data of *Figure 2C* and *Figure 3*). One might speculate that in vivo, and especially in the (pro)heterocyst, the affinity of HetR for HetL increases either due to a modification of HetR or to its interaction with another factor. HetR has been shown to be regulated by phosphorylation (*Valladares et al., 2016*; *Roumezi et al., 2019*), if this posttranslational modification occurs only in the developing cell it could explain a high affinity of HetR for HetL specifically in the heterocyst. In addition, the observation that the overexpression of *hetL* in vegetative cells, from *petE* or *rbcL* promoters, did not lead to their differentiation into heterocsyts is rather in favor of a mechanism promoting the association between HetR and HetL specifically in the heterocyst. Resolving the structure of HetR-HetL complex and following the behavior of HetL protein and PatS in vivo will provide more information about the dynamics and the nature of HetR complexes in the two cell types through the differentiation process.

A deletion mutant of *hetL* differentiates heterocysts as well as the wild-type strain (*Liu and Golden, 2002*). The possibility that *hetL* is not essential for heterocyst differentiation and that only its overexpression affects the differentiation process was one of the hypotheses proposed to explain the phenotype of the *hetL* mutant. The second hypothesis was functional redundancy based on a crosstalk between HetL and another PRP (*Liu and Golden, 2002*). The observation that at least one homolog of HetL (All4303) interacted with HetR, is rather in favor of functional redundancy. A global transcription study using RNA sequencing has shown that all4303 transcription is induced in response to nitrogen starvation (*Flaherty et al., 2011*). The *hetL* gene has also been reported in this study to be induced after nitrogen starvation, which is consistent with our data. The other PRPs coding genes whose transcription has also been reported to be regulated in this condition are reported in *Supplementary file 1*. At this stage of our investigation, we cannot rule out that homologs of HetL may also be part of the PatS signaling, but the fact that the genetic suppressor screen for suppression of PatS inhibition selected only *hetL* would rather point to it as the principal factor involved in the differentiation process (*Liu and Golden, 2002*). Because of the presence of multiple homologs, cross-complementation might occur when *hetL* happens to be mutated.

Genes encoding proteins containing tandem pentapeptide repeats are abundant in all cyanobacterial genomes, which render the study of their function by gene-deletion approaches unjustified. Interestingly, despite their high amino acid homology, their structures possess distinctive features that can help to investigate their function. Exploring these structural features and searching for potential binding-partners can be a successful strategy in their study. The gyrase-binding PRP is the first protein of this family whose partner was identified (*Tran et al., 2005*). PRPs from this family are largely conserved among bacteria, where number of them have been, based on their ability, proposed to provide resistance to quinolone-type antibiotics. Structural studies revealed the distribution of large contiguous patches of negative electrostatic potential resembling DNA which gave insights into their function as mimicking DNA structure for binding to gyrase (*Xiong et al., 2011*). The structure of HetL does not show such a large distribution of negative charges, but instead

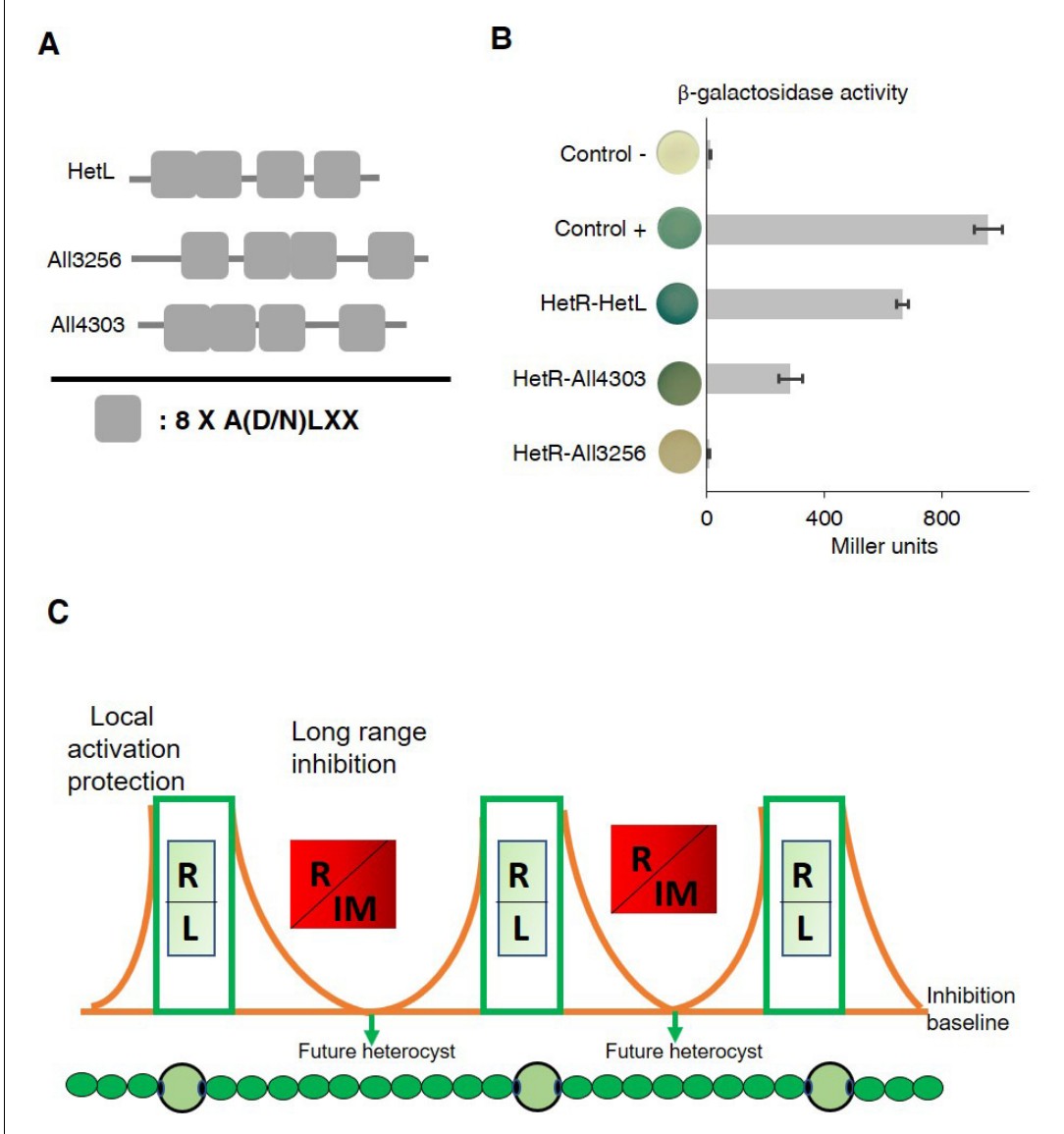

**Figure 7.** HetR interacts with a HetL homolog and patterning model including HetL. (**A**) HetL, All3256 or All4303 PR domains organization. (**B**) Bacterial two hybrid assay between HetR and HetL paralogs. BTH101 strain was transformed with pKT25-*hetR* and pUT18C-all4303 or pUT18C-all3256, β-galactosidase activities were measured as described in section 'Materials and methods' and were expressed in Miller units. Strains producing the T18 and T25 served as negative control. Strains producing T18-Zip and T25-Zip served as positive control. Error bars indicate standard deviation. The characteristics of the fusion proteins used in this assay are indicated below: HetR-HetL: T25-HetR/T18-HetL; HetR-All4303: T25-HetR/T18-All4303; HetR-All3256: T25-HetR/T18-All3256. (**C**) Schematic model integrating HetL in the patterning system of *Nostoc*. R: HetR, L: HetL or HetL homolog, IM: inhibiting morphogen (Providing from the processing of PatX, PatS or HetN). **Local activation/protection**: once activated, HetR activates the transcription of *hetR*, *hetL*, *patS/hetN*. The HetR/HetL network is favored. HetL provides immunity against the inhibiting morphogens produced in situ or entering the cell. HetR is active, heterocyst develops. **Long range inhibition**: the diffusion of the morphogens from both sides of the heterocyst creates an inhibition gradient. In the HetR/Morphogen network, HetR is inactive and the concentration of HetL is below the protective threshold. Differentiation is inhibited.

The online version of this article includes the following source data for figure 7:

**Source data 1.** Source data to *Figure 7B*.

contiguous patches of positively and negatively charged surfaces are distributed at the surface of the structure. These patches have been suggested to mediate the binding to unknown potential partners, other than DNA (*Ni et al., 2009*). In docking simulation analyses reported in this study, these charged surfaces especially those localized in face 2 and face 3 mediate the interaction with

the HetR-Hood domain. To the best of our knowledge, the interaction of HetL with HetR constitutes the second case of protein-protein complex identified for a PRP member. In *Nostoc* two other integral PRPs (PatL [*Liu and Wolk, 2011*] and FraF [*Merino-Puerto et al., 2013*]) and a pentapeptide repeat domain-containing protein (HglK, *Black et al., 1995*) are involved in cell differentiation, but their mechanism of action is not fully uncovered. Analyzing their interaction with their physiological partners will provide a better understanding of the role played by PRPs in the developmental program of *Nostoc*.

Genetic experiments showed that HetL interaction with HetR allows the differentiating cell to escape from PatS overproduction and from the addition to the culture medium of the (E)**RGSGR,** (H) **RGTGR** inhibiting peptides (*Figures 4* and *6*). These data suggest that HetL provides immunity for the developing cells all along the differentiation process. HetL might therefore be important for pattern initiation and establishment, exerted by PatX and PatS, and also for pattern maintenance mediated by HetN and PatS. Therefore, we propose that HetR can be engaged in two different networks depending on cell types: a HetR-HetL-positive network in the (pro)heterocyt as a consequence of *hetL* expression, and a negative HetR-inhibiting peptide complex taking place in the HetL-free vegetative cells. The HetR/HetL complex therefore enriches the 'local-activation and long-range inhibition model' with a third module that we propose to call 'local protection' since this complex allows the differentiating cell to become immune to self-inhibition (*Figure 7C*). An interesting particularity of patterning in *Nostoc* is that future heterocysts do not emerge as isolated cells. Rather, strings of 3–4 developing cells appear before resolving into a single proheterocyte. (*Wilcox et al., 1973*). These clusters of cells 'competent' for differentiation were suggested to be correlated to the stochastic fluctuation in *hetR* expression (*Corrales-Guerrero et al., 2015*). In addition, we propose that a stochastic variation of *hetL* expression among the string of competent cells might also be part of the resolving mechanism fixing the decision to differentiate in the cell that inherits enough HetL to be protected from self-inhibition. Microfluidic approaches probing multiple and integrated aspects of heterocyst formation along with mathematical models should shed light on how patterning emerges, resolves and persists in *Nostoc*. An interesting evolutionary aspect resulting from the conservation of PRPs coding-genes in cyanobacterial genomes is that the function of HetL described here might be accurate for other heterocyst-forming cyanobacteria. It is also likely that understanding all the parameters and factors governing heterocyst-pattern formation will be useful in uncovering the molecular mechanisms underlying patterning in more complex biological systems.

# Materials and methods

### Key resources table

| Reagent type (species) or resource | Designation | Source or reference | Identifiers | Additional information |
|---|---|---|---|---|
| Gene (*Nostoc* PCC 7120) | *hetR* | GenBank | Genomic sequence: NC_003272.1; Gene ID:1105936 | |
| Gene (*Nostoc* PCC 7120) | *patS* | GenBank | Gene ID:1105898 | |
| Gene (*Nostoc* PCC 7120) | *hetL* | GenBank | Gene ID:1107338 | |
| Gene (*Nostoc* PCC 7120) | all4303 | GenBank | Gene ID:1107905 | |
| Gene (*Nostoc* PCC 7120) | all3256 | GenBank | Gene ID:1106854 | |
| Strain, strain background (*Nostoc*) | Wild-type strain; WT | Pasteur Institute Collection | PCC 7120 | |
| Strain, strain background (*Nostoc*) | ΔhetR | DOI:10.1111/j.1365-2958.2005.04678.x | | *Nostoc* PCC 7120 deletion mutant of the *hetR* gene |

*Continued on next page*

*Continued*

| Reagent type (species) or resource | Designation | Source or reference | Identifiers | Additional information |
|---|---|---|---|---|
| Strain, strain background (*Escherichia coli*) | BL21DE3 | New England Biolabs | C2527 | competent cells for production of proteins |
| Strain, strain background (*Escherichia coli*) | BTH101 | DOI:10.1073/pnas.95.10.5752 | | competent cells for two hybrid assays |
| Antibody | Anti-histidine (mouse monoclonal) | Qiagen | Cat# 34660; RRID:AB_2619735 | (1:1000) |
| Recombinant DNA reagent | pRL1272 (plasmid) | DOI:10.1128/jb.170.3.1239-1244.1988 | | replicative in *Nostoc*, encoding resistance to erythromycin |
| Recombinant DNA reagent | pCSB270 (plasmid) | Other | | lab collection (pRL1272 with *petE* promoter) |
| Recombinant DNA reagent | pRL25T (plasmid) | DOI: 10.1016/j.resmic.2012.10.010 | | replicative in *Nostoc* encoding resistance to neomycin |
| Recombinant DNA reagent | pCSB265 (plasmid) | Other | | lab collection (pRL25T bearing the *petE* promoter) |
| Sequence-based reagent | P*hetP* fw (PCR primers) | Other | [6FAM]ATTTAGTGGTAAATTCTCTT | lab collection |
| Sequence-based reagent | P*hetP* rv (PCR primers) | Other | TGAGTTATACGCTATATCAA | lab collection |
| Peptide, recombinant protein | PatS-5 (synthesized peptide) | Genecust | RGSGR | desalted > 99% |
| peptide, recombinant protein | PatS-6 (synthesized peptide) | Genecust | ERGSGR | desalted > 99% |
| Peptide, recombinant protein | PatX-6 (synthesized peptide) | Genecust | HRGTGR | desalted > 99% |
| Commercial assay or kit | In-Fusion HD Cloning kit | Takara | 639646 | |
| Commercial assay or kit | GoScript Reverse transcriptase | Promega | A5000 | |
| Chemical compound, drug | HisTrap Excel | GE healthcare | GE17-3712-05 | |
| Chemical compound, drug | Disposable PD 10 Desalting Columns | GE healthcare | GE17-0851-01 | |
| Chemical compound, drug | DiDC competitor | Sigma-Aldrich | 81349 | |
| Software, algorithm | HADDOCK2.2 webserver | DOI: 10.1016/j.jmb.2015.09.014 | | http://milou.science.uu.nl/services/HADDOCK2.2/ |

## Strains, plasmids and primers

The strains, plasmids and primers used in this study are listed in *Supplementary file 1*. All the cyanobacterial strains are derivatives of *Nostoc* PCC 7120 (Pasteur Cyanobacterial Collection, https://www.pasteur.fr/fr/sante-publique/crbip/les-collections/collection-cyanobacteries-pcc).

## Growth conditions, conjugation and heterocyst induction

*Nostoc* and derivatives were grown in BG11 medium (*Rippka R et al., 1979*) at 30°C under continuous illumination (40 µE m$^{-2}$s$^{-1}$). When appropriate, media were supplemented with antibiotics at the following concentrations neomycin (50 µg mL$^{-1}$), erythromycin (200 µg mL$^{-1}$). Heterocyst formation was induced by transferring the exponentially growing cultures (OD 750 = 0.8) to BG110 (BG11 without sodium nitrate) medium. The presence of heterocysts was confirmed by microscopy.

Conjugation of *Nostoc* was performed as described in *Cai and Wolk, 1990*. Briefly, *E. coli* strains (bearing the replicative plasmid and the RP-4 conjugative plasmid) grown to exponential growth phase, were mixed to an exponentially grown *Nostoc* culture. The mixture was plated on BG11 plates and antibiotics were injected under the agar 24 hr later for plasmid selection.

## Plasmid construction

All the plasmids used in this study are listed in *Supplementary file 1*.

The strategy used for plasmid construction is briefly described below. All the recombinant plasmids were analyzed by sequencing.

### pXX1: pKT25-*hetR*

The open-reading frame of *hetR* gene (alr2339) was amplified using the *hetR* dh fw T25 and *hetR* dh rv primers and cloned into the PstI and EcoRI restriction sites of the pKT25 expression plasmid.

### pXX2: pKT25-*hetL*

The open-reading frame of *hetL* gene (all3740) was amplified using the *hetL* dh fw T25 and *hetL* dh rv primers and cloned into the PstI and EcoRI restriction sites of the pKT25 expression plasmid.

### pXX3: pKT25-*hetL*D151A

The pXX2 plasmid was used as template to substitute the D151 residue of HetL to Alanine. For this the primers Mut *hetL*D151A fw and Mut *hetL*D151A rv were used as primers in a megapriming PCR assay.

### pXX4: pUT18C-*hetR*

The open-reading frame of *hetR* gene (alr2339) was amplified using the *hetR* dh fw T18 and *hetR* dh rv primers and cloned into the PstI and EcoRI restriction sites of the pUT18C expression plasmid.

### pXX5: pUT18C-*hetR*R223W

The pXX4 plasmid was used as template to substitute the R223 residue of HetR to Tryptophan. For this, the primers Mut *hetR*R223W fw and Mut *hetR*R223W rv were used as primers in a megapriming PCR assay.

### pXX6: pUT18C-*hetL*

The open-reading frame of *hetL* gene (all3740) was amplified using the *hetL* dh fw T18 and *hetL* dh rv primers and cloned into the PstI and EcoRI restriction sites of the pUT18C expression plasmid.

### pXX7: pUT18C-*hetL*-RBS-*patS*

The open-reading frame of *patS* gene (asl2301) with extra 350 bp after the stop codon, was amplified using the RBS-*patS dh* fw T18 and RBS-*patS dh* rv T18 primers and cloned into the SalI and XhoI restriction sites of the pXX6 expression plasmid. The RBS is the one from the *petE* promoter.

### pXX8: pUT18C-*hetL*-RBS-*patS6*

The open-reading frame of *patS6* (encoding for PatS6 peptide: ERGSGR) with extra 350 bp after the stop codon, was amplified using the RBS-*patS6 dh* fw T18 and RBS-*patS6 dh* rv T18 infusion primers and cloned into the SalI restriction site of the pXX6 expression plasmid. The RBS is the one from the *petE* promoter.

### pXX9: *hetR*$_{hood}$-pUT18

The open-reading frame of *hetR*$_{hood}$ (encoding for HetR$_{hood}$ from Y215 to R296) was amplified using the *hetR*$_{hood}$ dh fw and *hetR*$_{hood}$ dh rv primers and cloned into the PstI and EcoRI restriction sites of the pUT18 expression plasmid.

### pXX10: pUT18C-all3256

The open-reading frame of all3256 gene was amplified using the all3256 dh fw T18 and all3256 dh rv primers and cloned into the PstI and EcoRI restriction sites of the pUT18C expression plasmid.

### pXX 11: pUT18C-all4303

The open-reading frame of all4303 gene was amplified using the all4303 dh fw T18 and all4303 dh rv primers and cloned into the PstI and EcoRI restriction sites of the pUT18C expression plasmid.

### pXX12: pET28a-*hetL*-his

The open-reading frame of *hetL* gene (all3740) was amplified using the *hetL* pET28 fw and *hetL* pET28 rv infusion primers and cloned into the XhoI and NcoI restriction sites of the pET28a expression plasmid using the In-Fusion technology (Takara In-Fusion HD Cloning kit).

### pXX13: pRL1272-P*petE-patS*

The open-reading frame of *patS* gene (asl2301) with extra 700 bp after the stop codon, was amplified using the *patS* pRL fw and *patS* pRL rv infusion primers and cloned into the BamHI restriction site of the pRL1272-P*petE* replicative plasmid in *Nostoc* (Takara In-Fusion HD Cloning kit). P*petE* stands for: *petE* promoter.

### pXX14: pRL25T-P*petE-hetL*

The open-reading frame of the *hetL* gene (all3740) was amplified using the *hetL* pRL fw and *hetL* pRL rv infusion primers and cloned into the BamHI restriction site of the pRL25T-P*petE* replicative plasmid in *Nostoc* (Takara In-Fusion HD Cloning kit).

### pXX15: pRL25T-P*petE-hetL*D151A

The open-reading frame of the *hetL* gene mutated to encode for a D151A substitution was amplified using the *hetL* pRL fw and *hetL* pRL rv infusion primers with pKT25-*hetL*D151A as template and cloned into the BamHI restriction site of the pRL25T-P*petE* replicative plasmid in *Nostoc* (Takara In-Fusion HD Cloning kit).

### pSC1: pRL25T-P*patS-hetL*

The promoter of the *patS* gene was amplified using P*patS* fw and P*patS* rv, the *hetL* coding region was amplified using *hetL* P*patS* fw and *hetL* P*patS* rv. *Nostoc* genomic DNA was used as template for both amplifications. The two amplicons were cloned into the BamHI restriction site of the pRL25T plasmid using the In-Fusion technology.

### pSC2: pRL25T-P*rbcL-hetL*

The promoter of the *rbcL* gene was amplified using P*rbcL* fw and P*rbcL* rv, the *hetL* coding region was amplified using *hetL* P*rbcL* fw and *hetL* P*rbcL* rv. *Nostoc* genomic DNA was used as template for both amplifications. The two amplicons were cloned into the BamHI restriction site of the pRL25T plasmid using the In-Fusion technology.

## Protein purification

For HetL purification, the BL21DE3 strain containing the pXX12 plasmid was grown until an optical density (OD 600 nm) of 0.6. The expression of *hetL* was induced by the addition of Isopropyl β-D-1-thiogalactopyranoside (IPTG, SIGMA) of 0.4 mM over night at 16°C. Cells were harvested at 8000 rpm at 4°C during 2 min. The pellet was re-suspended in 25 mL of lysis buffer (50 mM Tris HCl (pH 8), 0.3 M NaCl), and cells were disrupted using French press. After centrifugation at 8000 rpm for 30 min at 4 °C, the supernatant was loaded onto a HisTrap Excel column (GE healthcare) pre-equilibrated with lysis buffer containing 10 mM Imidazole. The column was rinsed with 10 mM and 35 mM Imidazole, both prepared in lysis buffer. Fractions were collected (in 200 mM Imidazole). The Imidazole was eliminated using disposable PD 10 desalting columns (GE Healthcare). The proteins were concentrated using Vivaspin columns (SIGMA) and quantified using the Bradford assay (SIGMA). HetR purification was undergone as previously described (*Hu et al., 2015*).

## Electrophoretic mobility shift assays (EMSA)

The promoter region of the *hetP* gene (alr2818) was obtained by PCR using P*hetP* fw and P*hetP* rv primers. The forward primer was modified at its 5' end by adding the 6-carboxyfluorescein (6-FAM) dye. Purified HetR (1 μM) and HetL (2–4 μM) proteins, were incubated with the *hetP* promoter (50 nM) in a buffer containing 10 mM Tris (pH 8), 150 mM potassium chloride, 500 nM EDTA, 0.1% Triton X-100, 12.5% glycerol, 1 mM dithiothreitol and 1 μg DiDC competitor (poly(2'-deoxyinosinic-2'-deoxycytidylic acid) sodium salt), at 4°C in dark for 30 min. The electrophoresis was performed at 250 volts for 60 min. The DNA was revealed using Typhoon FLA 9500 (GE Healthcare Life Sciences). The experiment was repeated three times with independent protein purifications and one representative result is shown.

## RNA preparation, reverse transcription, and quantitative real-time-PCR

RNAs were prepared using the Qiagen RNA extraction kit (Qiagen) following the manufacturer's instructions. An extra TURBO DNase (Invitrogen) digestion step was performed to eliminate the contaminating DNA. The RNA quality was assessed by tape station system (Agilent). RNAs were quantified spectrophotometrically at 260 nm (NanoDrop 1000; Thermo Fisher Scientific). For cDNA synthesis, 1 μg total RNA and 0.5 μg random primers (Promega) were used with the GoScript Reverse transcriptase (Promega) according to the manufacturer instructions. Quantitative real-time PCR (qPCR) analyses were performed on a CFX96 Real-Time System (Bio-Rad). The reaction volume was 15 μL and the final concentration of each primer was 0.5 μM. The qPCR cycling parameters were 95°C for 2 min, followed by 45 cycles of 95°C for 5 s, 55°C for 60 s. A final melting curve from 65°C to 95°C was added to determine the specificity of the amplification. To determine the amplification kinetics of each product, the fluorescence derived from the incorporation of BRYT Green Dye into the double-stranded PCR products was measured at the end of each cycle using the GoTaq qPCR Master Mix 2X Kit (Promega). The results were analysed using Bio-Rad CFX Maestro software, version 1.1 (Bio-Rad, France). The RNA 16S gene was used as a reference for normalization. All measurements were carried out in triplicate and a biological duplicate was performed for each point. The amplification efficiencies of each primer pairs were 80% to 100%. All the primer pairs used for qPCR are reported in *Supplementary file 1*.

## Bacterial two hybrid assays

For each construct, the studied proteins were fused to the carboxy terminus of the T25 or T18 domain of CyA, except that HetRhood was fused to the N-terminus of the T18 domain. Bacterial two-hybrid assays were performed following the procedure described by *Karimova et al., 1998*. Briefly, after co-transforming the BTH101 strain with the two plasmids expressing the T18- and T25-fusions, LB plates containing ampicillin and kanamycin were incubated at 30° C for 2 days. For each assay, 10 independent colonies were inoculated in 3 ml of LB medium supplemented with ampicillin, kanamycin and 0.5 mM IPTG, and incubated at 30°C overnight. ß-galactosidase activity was determined as previously described (*Zubay et al., 1972*). The values presented are means of three independent assays.

## Sodium dodecyl sulfate–polyacrylamide gel electrophoresis (SDS-PAGE)

Proteins were fractionated by performing SDS-PAGE (4–20%) stained with Coomassie blue (Euromedex, Souffelweyrshim, France). For immunoblot analysis, the proteins were transferred to nitrocellulose membranes before being revealed with anti-Histidine monoclonal antibodies (Qiagen). Immune complexes were detected with anti-rabbit peroxidase-conjugated secondary antibodies (Promega) and enhanced chemiluminescence reagents (Pierce, Illkich, France).

## Synthetic peptides

PatS-5 (RGSGR), Pat-6 (ERGSGR) and PatX-6 (HRGTGR) peptides were synthesized by Genecust (https://www.genecust.com/en/).

## Isothermal titration calorimetry (ITC)

ITC was performed to demonstrate the interaction of HetR and HetL with PatS peptides. The working buffer for both proteins and peptides was PBS pH 7.4 to avoid buffer mismatch. The

experiments were performed at 25℃ using the MicroCal PEAQ-ITC (Malvern UK) with 19 injections, first with an initial injection of 0.4 µL followed by 18 injections of 2 µL. The protein ligands were in the cell and the peptide analytes were in the syringe. The reaction was performed with a constant stirring speed of 750 rpm, each injection lasted for 4 s with a 150 s space between each injection. A constant heat control (offset) was removed from the raw data to account for heat dilution before integration. The data were fitted using a 'One Set of Sites' model in the PEAQ-ITC Analysis Software. The experiment was repeated two times with independent protein purifications and one representative result is shown.

### Bio-layer interferometry assays

The BLi machine (BLItz) from FortéBio was used to perform biolayer interferometry to determine the interaction between HetR and HetL. 2.2 µM of biotinylated HetR was loaded onto streptavidin biosensors in PBS. A 30 s baseline in PBS was performed before a 120 s association step with various concentrations of HetL at 2.5, 5, 10 and 20 µM followed by a 120 s dissociation step.

To determine if the PatS-5 and PatS-6 peptides compete with HetL for HetR binding, 10 µM of HetL was incubated five mins with different concentrations of PatS at 0.05, 0.1, 0.5 and 2.5 µM before adding HetL to the bound HetR. All bindings were performed in triplicate, the dissociation constant was obtained using the BLItz Pro Data Analysis software using a global 1:1 fit model.

### Molecular docking simulation

The available atomic coordinates of the hood domain of *Nostoc* HetR (PDB ID: 4YNL) and HetL (PDB ID: 3DU1) were used as templates for molecular docking simulations. Molecular docking study of the HetR with HetL was performed using the HADDOCK2.2 webserver (http://milou.science.uu.nl/services/HADDOCK2.2/) (*van Zundert et al., 2016*). The goal was to identify critical residues involved in HetR-HetL complex formation. To define Haddock run restraints, the surface exposed residues of both HetR and HetL were considered as active residues directly involved in the interaction. For both structures, the surface exposed residues were selected manually using PyMol (*Supplementary file 1*). Two independent docking simulations were performed. For each run, 10 clusters were generated, and classified based on their HADDOCK score. The best model of each cluster was analyzed by PDBePISA to explore their binding interfaces.

## Acknowledgements

The authors thank Yann Denis from the transcriptomic plateform (CNRS, IMM) for qRT-PCR analyses, and Aurélia Battesti and Maryline Foglino for helpful discussions.

## Additional information

### Funding

| Funder | Author |
| --- | --- |
| Centre National de la Recherche Scientifique | Xiaomei Xu |

The funders had no role in study design, data collection and interpretation, or the decision to submit the work for publication.

### Author contributions

Xiaomei Xu, Data curation, Formal analysis; Véronique Risoul, Stéphanie Champ, Formal analysis; Deborah Byrne, Data curation, Software, Methodology, Writing - review and editing; Badreddine Douzi, Conceptualization, Data curation, Software, Formal analysis; Amel Latifi, Conceptualization, Data curation, Supervision, Funding acquisition, Validation, Investigation, Methodology, Writing - original draft, Project administration, Writing - review and editing

### Author ORCIDs

Amel Latifi (iD) https://orcid.org/0000-0002-0776-7349

Decision letter and Author response
Decision letter https://doi.org/10.7554/eLife.59190.sa1
Author response https://doi.org/10.7554/eLife.59190.sa2

## Additional files

### Supplementary files
• Supplementary file 1. List of the strains and plasmids used in the study. Sequences of the primers used. Supplementary Table 1. PRPs encoding genes in the genome of *Nostoc* The sequences were retrieved from the Microscope Mage database. The genes located on *Nostoc* plasmids are marked by an asterisk. The functional domains were analyzed by Pfamscan. The cellular localization was deduced from the presence of transmembrane domain, signal peptide and lipopeptide using TMhmm, SignalIP and LipoP softwires. For gene expression, the reads values are those reported in *Flaherty et al., 2011*. Supplementary Table 2. List of active residues used as input for HetRhood: HetL docking simulations

• Transparent reporting form

### Data availability
All data generated or analysed during this study are included in the manuscript and supporting files. Source data files are provided for bacterial two hybrid analysis, qRT-PCR and heterocyst intervals and percentages.

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
