## [Decision Letter]

**Acceptance summary:**

This paper reveals a key aspect of cellular differentiation in the one-dimensional patterning of cyanobacterial heterocysts. The differentiating heterocyst produces a morphogen, PatS, whose gradient emanating from the source suppresses the differentiation of its neighbors and establishes spacing of heterocysts every 10-12 cells. Here Xu et al. describe how the PatS-producing cell escapes suppression of differentiation: another protein, HetL, also expressed in the nascent heterocyst, directly competes with PatS for binding to the transcription factor HetR, blocking the developmental suppression activity of PatS.

**Decision letter after peer review:**

Thank you for submitting your article "HetL provides immunity to HetR against PatS inhibition, and promotes patterning in the cyanobacterium *Nostoc* PCC 7120" for consideration by *eLife*. Your article has been reviewed by three peer reviewers, including Susan Golden as the Reviewing Editor and Reviewer #1, and the evaluation has been overseen by Gisela Storz as the Senior Editor. The following individual involved in review of your submission has agreed to reveal their identity: Conrad W Mullineaux (Reviewer #2).

The reviewers have discussed the reviews with one another and the Reviewing Editor has drafted this decision to help you prepare a revised submission.

Summary:

The authors present a compelling study that goes a long way towards explaining how PatS/HetR/HetL can serve as a Turing-type reaction-diffusion system to control pattern formation in differentiating filaments of the cyanobacterium *Nostoc* (*Anabaena*). Heterocyst formation in *Nostoc* produces a one-dimensional pattern; this minimal system has always looked like a good model for a rigorous understanding of biological pattern formation and should have broad appeal for the readers of *eLife*. Although the study doesn't provide rigorous quantitative understanding of the model, it defines the key players in the system and explains qualitatively how they act. The work shows that a pentapeptide repeat protein called HetL interacts with the transcriptional regulator HetR in competition with the PatS peptide. PatS inhibits HetR interaction with promoter targets, and HetL engagement with HetR blocks PatS-mediated inhibition. In this way, the differentiated cells that produce PatS are immune to inhibition, whereas surrounding vegetative cells are inhibited by a gradient of PatS (as previously demonstrated). The authors provide data in the form of protein-protein and protein-DNA interactions, structural modelling, gene expression, and genetic approaches that support a mechanistic model in which HetL and PatS interact with the same surface of HetR. Additionally, the authors identify a second protein, All4303, that is homologous to, and may serve a redundant function as that of HetL.

Revisions:

One substantive scientific point: the schematic model in Figure 7C doesn't make sense, because it shows diffusion of the inhibiting morphogen away from the heterocyst in one direction only (towards the right, as shown in the scheme). All available evidence indicates that *Nostoc* cells are equally open for molecular exchange at both poles. A much more likely scenario would show the morphogen concentration declining symmetrically either side of each heterocyst. Indeed that was what was shown experimentally by Corrales-Guerrero et al., 2013, who found the highest concentrations of PatS in the vegetative cells either side of each heterocyst.

Authors don't address the question of how the PatS peptide diffuses from cell to cell, and something on that point in the Discussion section would help to give readers a more complete picture of the system. Circumstantial evidence suggests that PatS diffuses via the "septal junctions" (Mariscal et al., 2016). The septal junction structures have recently been very nicely characterised by cryo-electron tomography (Weiss et al., 2019). The observed dimensions of the intercellular channels may explain why the PatS peptide can diffuse from cell to cell, but HetL can't (which is a key requirement for the authors' model to work).

It would be good to have a title that captures the broader significance of the work. Maybe something like: "HetL, HetR and PatS form a reaction-diffusion system for pattern formation in the filamentous cyanobacterium *Nostoc*"

The impact statement doesn't do the work justice, as it emphasizes details. The work describes elements of a Turing-type reaction-diffusion system to control pattern formation in differentiating cyanobacterial filaments.

---

## [Author Response]

Revisions:One substantive scientific point: the schematic model in Figure 7C doesn't make sense, because it shows diffusion of the inhibiting morphogen away from the heterocyst in one direction only (towards the right, as shown in the scheme). All available evidence indicates that Nostoc cells are equally open for molecular exchange at both poles. A much more likely scenario would show the morphogen concentration declining symmetrically either side of each heterocyst. Indeed that was what was shown experimentally by Corrales-Guerrero et al., 2013, who found the highest concentrations of PatS in the vegetative cells either side of each heterocyst.

We appreciate the concern. In the original Figure 7, we decided to present the diffusion of PatS from only one part of the differentiating cell which is incomplete and confusing for a reader non-familiar with the differentiation process. In the revised Figure 7, we corrected this point by drawing the diffusion of PatS from both sides of the heterocyst.

Authors don't address the question of how the PatS peptide diffuses from cell to cell, and something on that point in the Discussion section would help to give readers a more complete picture of the system. Circumstantial evidence suggests that PatS diffuses via the "septal junctions" (Mariscal et al., 2016). The septal junction structures have recently been very nicely characterised by cryo-electron tomography (Weiss et al., 2019). The observed dimensions of the intercellular channels may explain why the PatS peptide can diffuse from cell to cell, but HetL can't (which is a key requirement for the authors' model to work).

We agree that the point raised here is important to address to give the reader a better understanding of the action of the inhibiting peptides. We also agree that discussing the structure of the septal junctions strengthens our model of the mechanism of HetL. We have addressed this point in the Discussion section.

It would be good to have a title that captures the broader significance of the work. Maybe something like: "HetL, HetR and PatS form a reaction-diffusion system for pattern formation in the filamentous cyanobacterium Nostoc"

It was important for us to choose a title which highlights the novelty that our study brings in relation to the state of art in the field. Nonetheless, we understand the importance of broad-significance titles and therefore decided to adopt the proposed one.

The impact statement doesn't do the work justice, as it emphasizes details. The work describes elements of a Turing-type reaction-diffusion system to control pattern formation in differentiating cyanobacterial filaments.

We agree. The impact statement has been changed following the suggestion.